# Drug Utilization and Measurement of Medication Adherence: A Real World Study of Psoriasis in Italy

**DOI:** 10.3390/pharmaceutics15122647

**Published:** 2023-11-21

**Authors:** Sara Mucherino, Concetta Rafaniello, Marianna Serino, Alessia Zinzi, Ugo Trama, Annalisa Capuano, Enrica Menditto, Valentina Orlando

**Affiliations:** 1Center of Pharmacoeconomics and Drug Utilization Research (CIRFF), Department of Pharmacy, University of Naples Federico II, 80131 Naples, Italy; sara.mucherino@unina.it (S.M.); marianna.serino@unina.it (M.S.); enrica.menditto@unina.it (E.M.); 2Department of Experimental Medicine—Section of Pharmacology “L. Donatelli”, University of Campania “Luigi Vanvitelli”, Via Costantinopoli 16, 80138 Naples, Italy; concetta.rafaniello@unicampania.it (C.R.); alessia.zinzi@unicampania.it (A.Z.); annalisa.capuano@unicampania.it (A.C.); 3Regional Pharmaceutical Unit, Campania Region, 80143 Naples, Italy; ugo.trama@regione.campania.it; 4HealthCare Datalab, Campania Region, 80143 Naples, Italy

**Keywords:** drug utilization, medication adherence, real world evidence, psoriasis, adherence measurement

## Abstract

Exceptional advances have been made with systemic treatment for psoriasis (PSO). However, that disease still represents a heavy burden in terms of impact on healthcare systems worldwide. This study comprehensively assesses medication adherence in a real world setting in Italy across all phases—initiation, implementation, and persistence—of PSO therapies. By distinguishing between switches and swaps, it provides unique insights into the patient’s own approach to prescribed therapy as well as clinical decision-making processes, enhancing our understanding of medication adherence and discontinuation in a real world daily setting. The study’s refined methodology for assessing persistence, considering variations in refill gaps and complex dosing regimens, shows that anti-interleukin (IL) therapies are associated with longer periods of adherence compared with other available therapeutic strategies. Among the selected drugs, ixekizumab and secukinumab were the ones with higher rate of treatment adherence at the expense of anti-TNF-α and anti-PDE4 agents. Notably, patients who opt for swaps are approximately 2.8 times more likely to discontinue their PSO therapy within one year. These findings carry practical implications for optimizing medication adherence, including tailored patient counseling, monitoring, and therapeutic adjustments, highlighting the need for a comprehensive and patient-centered approach to managing these conditions.

## 1. Introduction

Psoriasis, a chronic inflammatory skin condition, follows a relapsing–remitting course that significantly impacts the quality of life for those affected. In Europe, psoriasis prevalence varies among regions, affecting approximately 2–4% of the population in Western countries [1]. Specifically, in Italy, it is estimated that around 2.8% of individuals suffer from psoriasis [2,3]. Furthermore, there is a noteworthy trend in Italy of an increasing proportion of psoriasis cases among the elderly population, with 15% of patients aged 65 or older presenting with moderate to severe disease [4]. This demographic shift emphasizes the urgent need to address psoriasis in older individuals and develop targeted treatments and management strategies to enhance their health outcomes.

Over time, it has become increasingly evident that individuals with psoriasis often express substantial dissatisfaction with conventional medications [5,6,7,8], resulting in encountering significant adherence challenges [9]. In response to this scenario, the field of pharmacological treatment for psoriasis has witnessed remarkable advancements in the past decade with the introduction of biologic drugs and small molecules that have revolutionized the management of moderate-to-severe psoriasis [10]. The primary advantage of these biologic therapies lies in their more selective mechanisms of action, leading to improved health outcomes and reduced levels of therapeutic ineffectiveness [11,12]. These biologic drugs are categorized based on their mechanisms of action, including anti-tumor necrosis factor (TNF), anti-interleukin (IL)-12/23, anti-IL-17, and anti-IL-23 agents [13,14].

While all these biologic drugs, including small molecules, are highly effective and exhibit a favorable risk–benefit profile, variations in efficacy and adherence levels to therapy may exist. Recent studies have shown that overall adherence levels to biologic drugs are superior to conventional medications [15]. However, specific evidence suggests that adherence to biologic drugs can still fall short of optimal levels [13]. While some recent studies have assessed adherence to specific biologic drugs [16], there is currently a gap in knowledge concerning real world adherence levels across all classes of biologic drugs used for the treatment of psoriasis (PSO).

Therefore, the principal objective of this study is to comprehensively evaluate adherence to all PSO biologic treatments in a real world context by assessing the phases of the adherence process: initiation, implementation, and persistence with biologic therapy. Through this effort, we aim to offer valuable insights into the challenges and opportunities associated with psoriasis treatment using biologic drugs, with the ultimate goal of enhancing patient outcomes and the effectiveness of therapeutic interventions.

## 2. Materials and Methods

### 2.1. Data Source

A retrospective observational study was performed according to the Strengthening the Reporting of Observational Studies in Epidemiology guidelines [17] using as data source the Collection of Treatment Plans of Campania Region population, Southern Italian Region of about 6 million inhabitants (10% nationality representation). This data collection operates as a web platform for specialized facilities and public and private pharmacies within the Local Health Units (LHUs) of the Campania Region. It includes treatment plans providing information on diagnosis (with a focus on diagnosis codes for dermatologic pathologies according to the International Classification of Diseases, Ninth Revision, Clinical Modification, or ICD9-CM) and medication details (with a focus on Anatomical Therapeutic Chemical, or ATC, codes related to active substances prescribed, pharmaceutical form, dose, treatment cycle, and reimbursement form for psoriatic treatments). We integrated this database with the Campania Region DataBase (CaReDB), a health-electronic data warehouse previously validated in various studies [18,19,20]. CaReDB provided demographic information on all residents covered by the Regional Health System, including age and gender. These two databases were linked through a unique, encrypted, and anonymous identifier to ensure patient privacy protection.

### 2.2. Study Population

Patients who received a diagnosis of psoriasis (PSO) were recruited between 2017 and 2019. A graphical explanation of the study design can be found in Appendix A. Inclusion criteria for the selection of the final PSO cohort were as follows: (i) patients with a prescription of a therapeutic plan with an ATC code for PSO treatment (study drugs listed in Appendix A); (ii) patients with a prescription of a therapeutic plan with a PSO diagnosis (ICD9-CM PSO codes 680-709, 696.xx); (iii) patients with no prescription of any PSO drug in the year preceding the recruitment period (a 1-year washout from 2017 to 2018). Exclusion criteria were also applied for the selection of the study population, such as: (i) patients diagnosed with a non-dermatological condition during the entire study period, namely, rheumatoid arthritis (ICD9-CM code 714), ankylosing spondylitis (ICD9-CM code 720.0), Crohn’s disease (ICD9-CM code 555), and ulcerative colitis (ICD9-CM code 556); (ii) patients with less than 1-year follow-up data available; (iii) a non-representative sample of patients treated with PSO drug treatment, such as certolizumab, with only one patient (n = 1), rendering it statistically insignificant within the broader category of patients receiving those drugs. Please refer to Appendix A for a flowchart.

### 2.3. Adherence Measurement

International guidelines for the treatment of psoriasis recommend a two-phase pharmacological therapy, comprising an induction phase and a maintenance phase [21]. To assess adherence to each prescribed medication, a specific time duration has been allocated for each phase. The coverage period for each phase has been determined based on the quantity of tablets or syringes provided with each medication. A detailed description of the entire process is available in Appendix A. Medication adherence estimates were performed according the EMERGE guidelines on adherence developed under the umbrella of the European Ascertaining Barriers for Compliance (ABC) project. This taxonomy defines adherence as the process by which patients take their prescribed medications and comprises three essential components: (i) initiation; (ii) implementation; (iii) discontinuation [22,23,24]. The process starts with initiation when the patient takes the first dose of a prescribed medication. The process continues with the implementation of the dosing regimen, defined as the extent to which a patient’s actual dosing corresponds to the prescribed dosing regimen, from initiation until the last dose is taken. Discontinuation marks the end of therapy, when the next dose to be taken is omitted and no more doses are taken thereafter [22,23,24]. We assessed the three phases of the adherence process as follows: (i) initiation was determined by the discrepancy between the drug prescribed in the treatment plans and the drug actually dispensed; (ii) the implementation phase was evaluated through the estimation of switching (i.e., a change in the prescription of the index drug to another drug of a different drug class) and swapping (i.e., a change in the prescription of the index drug to another drug within the same drug class) rates. Switch and swap rates were calculated for incident patients and were not considered as interruptions of PSO therapy (see Appendix A); (iii) the discontinuation phase was evaluated by estimating the 1-year persistence to the index treatment, measured by the time gap between two successive drug dispensations. Patients were considered non-persistent if the gap between two refills exceeded 30 days (with a grace period). Sensitivity analyses were conducted using refill gap thresholds of 30, 60, and 90 days. The duration of medication supplied was determined based on the quantity of pills, syringes, and packages. Medication persistence was assessed at the drug class level. Patients were censored if they surpassed the allowed gap without obtaining a new prescription or if they reached the end of the study period (provided they had maintained persistence throughout the follow-up period). Non-persistent users were categorized as those who recommenced therapy after a period of discontinuation or those who simply discontinued treatment.

### 2.4. Covariates

Demographic characteristics, such as age and gender, of study population were collected and stratified according to the sub-cohorts of index PSO treatments: anti-PDE4 agents (apremilast), anti-TNF-α agents (etanercept, adalimumab), anti-IL-12/23 agents (ustekinumab), and anti-IL-17 agents (secukinumab, ixekizumab). The assessment of polypharmacy regimens was based on three categories: “excessive polypharmacy” for the prescription of ≥10 drugs per day; “polypharmacy” for the prescription of 5 to 9 drugs per day; “no-polypharmacy” for the concomitant use of ≤4 drugs per day [25].

### 2.5. Statistical Analyses

Descriptive statistics were employed to assess the baseline characteristics of the study population. Quantitative variables were presented as means with standard deviations, while categorical variables were expressed as counts and percentages. Persistence rates were determined using the Kaplan–Meier method, and statistical differences between survival curves were assessed using the log-rank test. Logistic regression models were utilized to estimate the likelihood of non-persistence over a 1-year period following PSO treatment initiation and to evaluate the factors influencing persistence. Odds ratios (ORs) and 95% confidence intervals (95% CI) were computed to assess both crude and adjusted associations for relevant predictors, with a significance threshold set at *p* < 0.05. Data management was performed with Microsoft SQL server (V.2018, Microsoft, Redmond, WA, USA), and all analyses were performed with SPSS software for Windows (V.17.1, SPSS) and platform R (V.3.6, The R Formulation for Statistical Computing, Vienna, Austria).

### 2.6. Ethical Considerations

Study research protocol adhered to the tenets of the Declaration of Helsinki 1975 and its later amendments. Permission to use anonymized data for this study was granted to the researchers of the Centro di Ricerca in Farmacoeconomia e Farmacoutilizzazione (CIRFF) by the governance board of Unità del Farmaco della Regione Campania. The research did not involve a clinical study, and all patients’ data were fully anonymized and were analyzed retrospectively. For this type of study, formal consent was not required according to the current national established by the Italian Medicines Agency, and according to the Italian Data Protection Authority, neither ethical committee approval nor informed consent was required for our study [26].

## 3. Results

Overall, 25,138 patients were prescribed with medications, which included anti-PDE4, anti-TNF-α, anti-IL-12/23, and anti-IL-17 agents. Out of these patients, 15,363 had a diagnosis of psoriasis (PSO). Further analysis identified 1170 incident cases of PSO. After applying the aforementioned exclusion criteria, we established a final incident cohort comprising 801 PSO patients (refer to Appendix A). Within this cohort, 60.5% of patients were male, with an average age of approximately 50 years. Among the identified index PSO treatments, secukinumab (n = 183, 22.8%) and apremilast (n = 165, 20.6%) were the most commonly prescribed, while etanercept (n = 56, 7.0%) was the least frequently prescribed. Notably, PSO patients treated with apremilast had a higher average age (60.9 ± 12.9 years) and more complex treatment regimens, with 60% of these patients receiving more than 10 drugs per day. In contrast, patients with less complex treatment regimens, who did not receive polypharmacy, were more likely to initiate PSO treatment with etanercept (39.3%) or secukinumab (37.7%) (see Table 1).

Estimates of medication adherence for each PSO drug treatment indicated low levels of adherence (48.7%) across all phases of adherence (as detailed in Table 2). Notably, the initiation phase estimation revealed that 7% of patients did not initiate the first prescribed therapy, with a particularly high rate of non-initiation observed among patients prescribed anti-PDE4 medications (e.g., apremilast), with 15.4% of these patients failing to initiate therapy. The implementation phase estimation showed that swap rates (13.1%) exceeded switch rates (1.5%). Patients who initiated treatment with etanercept were most likely to swap therapies (33.9%), with the most common swaps being to ustekinumab (42.1%) and secukinumab (26.3%) (as depicted in Figure 1). In contrast, lower levels of swapping were observed among patients treated with anti-IL-17 agents such as secukinumab (9.8%) and ixekizumab (7.8%), and those who did swap were more likely to change therapy to ustekinumab and guselkumab. Overall, among patients who swapped from anti-PDE4, anti-TNF-α, and anti-IL-12/23 agents, most opted to swap to anti-IL-17 agents (e.g., ixekizumab) (Appendix A). The analysis of the discontinuation phase, which includes persistence, revealed that patients treated with anti-IL-17 agents demonstrated higher levels of persistence on therapy. This was evident in the case of a 30-day grace period, with persistence rates for ustekinumab at 32.3%, 64.5% for a 60-day gap, and 82.3% for a 90-day gap. Similarly, for secukinumab, the rates were 60.1%, 73.8%, and 77.6% for the respective gap durations. In the case of ixekizumab, the persistence rates were 47.5%, 62.4%, and 68.1% for the same gap durations. These results are presented in Table 2 and illustrated in the Kaplan–Meier curve (Figure 2). Conversely, as observed in both Table 2 and Figure 2, patients treated with apremilast exhibited lower levels of persistence to therapy, with rates of 47.3% for a 30-day gap, 56.4% for a 60-day gap, and 60% for a 90-day gap. Additional Kaplan–Meier curves can be found in Appendix A.

Logistic regression analysis revealed factors associated with non-persistence to PSO agents at the one year mark following treatment initiation. Importantly, the analysis showed a statistically significant association of treatment swaps as a predictor of non-persistence in PSO therapy (adjusted odds ratio 95% confidence interval: 2.835) (Table 3).

## 4. Discussion

The therapeutic options for managing psoriasis have significantly expanded in recent years, providing clinicians with a wide array of systemic oral drugs and various biologic agents with distinct mechanisms of action [27]. This study introduces an innovative methodology in the realm of adherence, being the first in scientific literature to comprehensively assess the complex issue of psoriasis treatments by separately evaluating all three pivotal phases of this therapeutic approach: initiation, implementation, and discontinuation. This approach aligns with the stringent criteria set forth by the EMERGE guidelines, offering a comprehensive insight into medication adherence patterns for psoriatic conditions.

To be specific, our analysis was conducted on an entire regional population, selecting 801 patients with prescriptions for PSO treatment based on ATC codes between 1 January 2017, and 31 December 2019. The study population exhibited a higher representation of male individuals (60.5%), with a mean age of 49.2 (±16.3). While these findings represent approximately 10% of the Italian population, they are consistent with other studies addressing the same topic conducted in different settings [13,15,28].

Within this population, apremilast, along with secukinumab, emerged as the second most prescribed medication for psoriasis patients. Notably, apremilast distinguishes itself as a non-biologic agent, but rather a synthetic product, offering an oral therapy option that could potentially result in higher patient compliance and therapy persistence. Moreover, in line with recent European guidelines [28], apremilast, a small molecular inhibitor of phosphodiesterase 4 (PDE4), is recommended as a second-line treatment. This recommendation is particularly pertinent when the oral route is preferred, or when “conventional” systemic agents have proven ineffective, are contraindicated, or are poorly tolerated. Moreover, apremilast is recommended for patients with concurrent conditions like psoriatic arthritis and those with pre-existing malignancies [29,30].

In the context of the implementation phase, our study constitutes a noteworthy contribution to the field, as it marks the first analysis of therapy changes that distinguishes between switches and swaps, often considered jointly in the assessment of therapy changes. This distinction entails precisely measuring therapeutic switches, wherein one drug is replaced by another with the same mechanism of action (e.g., substituting one anti-TNF drug with another anti-TNF drug as a second-line treatment), and therapeutic swaps, which involve replacing one drug with another that has a different mechanism of action (e.g., exchanging an anti-TNF drug with a different biologic drug utilizing a distinct mechanism of action). For patients dealing with psoriasis, the practice of changing therapies is quite common. Recent clinical evidence has revealed that therapeutic swaps, specifically, can lead to additional improvements that may have a positive impact on patients’ quality of life. It is worth noting that much of the existing clinical evidence on medication changes primarily focuses on switches while overlooking the intricacies associated with therapeutic swaps [16,31,32,33,34]. Our dataset revealed that, overall, 93% of patients initiated the first prescribed therapy. However, patients who commenced treatment with etanercept exhibited a higher rate of therapy changes, both in terms of switches (3.6%) and, notably, swaps (33.9%). Furthermore, apremilast, despite being recommended for comorbid patients, showed a higher rate of swaps (14.5%). In contrast, patients who initiated treatment with ixekizumab and secukinumab demonstrated the lowest rates of therapy changes (ixekizumab: switch 1.4%, swap 7.8%; secukinumab: switch 2.7%, swap 9.8%).

These findings seem to align with previous literature, highlighting low rates of therapy switches [31,32,33,34]. Specifically, in the Italian context, Giometto et al. (2022) reported a low overall switch rate (20%) with anti-IL drugs [30]. Their analysis of administrative databases in Tuscany aimed to evaluate the pattern of biologic drug use for psoriasis and revealed a higher persistence to newer biologic drugs compared to anti-TNF drugs (including etanercept and adalimumab). These findings align with observations in an American real world data setting, as evidenced by Feldman et al. in 2021. Feldman’s study highlighted even lower persistence with anti-TNF agents, considering longer follow-up periods at 12, 18, and 24 months. The observed low persistence with apremilast is consistent with previous findings as described by Feldman SR and others [31,33]. These findings suggest multiple factors contributing to poor persistence, including potential drug inefficiency, loss of response over time, and the occurrence of adverse events [27,28]. It is worth noting that these factors, particularly the long-term loss of drug efficacy, often lead to therapy discontinuation in patients with psoriasis [35]. Furthermore, our approach to assessing persistence incorporates additional details not commonly found in the literature [36]. Specifically, we employed a methodology that accounts for variations in the refill gap between doses, considering the complex therapeutic regimen of these drugs, which involves induction doses followed by maintenance with different time intervals. This complexity in the treatment regimen was evaluated by varying the sensitivity of coverage days, and the results consistently show that the use of anti-IL therapies is associated with extended treatment persistence. This reaffirms the notion that anti-IL therapies are linked to longer periods of adherence, which holds significant clinical relevance in the management of PSO patients. In the discontinuation phase, our study demonstrates that patients receiving anti-IL therapies exhibit higher levels of persistence compared to other pharmacological treatments for PSO, confirming existing literature [11,13,16,37,38,39]. Furthermore, the higher persistence observed with anti-IL agents compared to anti-TNF agents could be interpreted in the context of recommended dosing schedules as outlined in their Summary of Product Characteristics (SmPC). Patients with psoriasis may be more inclined to adhere to drugs administered with longer intervals, both for induction and maintenance phases.

Finally, we conducted a multivariate analysis to identify potential predictor factors of non-persistence to PSO agents at the 1-year mark post-initiation. This supplementary analysis revealed that, among various factors examined, therapeutic swap emerged as the sole significant predictor, demonstrating a substantial likelihood (OR 2.8; 95% CI 1.774–4.529; *p* < 0.001) of therapy discontinuation, in contrast to other factors. These results are particularly intriguing as they underscore the critical role of distinguishing between therapeutic switch and therapeutic swap when considering treatment changes. This analysis corroborates the notion that employing a swap strategy is more favorable than opting for a simple switch in terms of therapy retention. It emphasizes that the choice of changing to another drug within the same drug class, as opposed to switching to a different class altogether, has a notable impact on the persistence of PSO therapy. This distinction could guide clinical decision making to improve patient outcomes and treatment effectiveness.

This study had several limitations that warrant consideration. Although it offers valuable insights into PSO drug usage patterns, especially concerning adherence, its scope is primarily confined to the Italian population. Consequently, the generalizability of our findings to other regions or healthcare systems may be limited, as patient behaviors and treatment patterns can differ significantly among diverse populations. Furthermore, the dataset lacks information regarding disease severity and staging, potentially influencing our results. Being based on retrospective observational data, this study may be susceptible to limitations, such as incomplete or missing information, potential biases, and the inability to establish causal relationships. While our analysis of therapy changes, particularly swaps, provides valuable insights, it does not elucidate the underlying reasons for these changes. A comprehensive understanding of the factors driving therapy alterations could offer a more holistic perspective on medication adherence, benefiting both clinicians and patients. In this regard, we are planning to further investigate the safety profile of the studied drugs in a real world setting.

Notwithstanding the aforementioned limitations, this study offers a comprehensive examination of medication adherence, covering all three critical phases: initiation, implementation, and persistence. The findings present valuable insights of great interest to the scientific and clinical community, providing a better understanding of PSO treatment adherence in a real world context. A notable feature of this study is its distinction between therapeutic switches and swaps, which contributes unique insights into patient experiences and clinical decision-making processes. This differentiation enhances our comprehension of medication adherence and discontinuation. Moreover, in the context of the discontinuation phase, the study employs a refined approach to assess persistence. This methodology accommodates variations in the refill gap between doses, accounting for the complexities in dosing regimens associated with these drugs. These regimens encompass both induction and maintenance doses with varying coverage periods.

## 5. Conclusions

In conclusion, this study stands as a robust and comprehensive exploration of medication adherence in the realm of psoriasis (PSO) treatment, encompassing all pivotal phases: initiation, implementation, and persistence. By examining patterns of therapy changes and making a clear distinction between switches and swaps, it provides unparalleled insights into the patient experiences and the decision-making processes of healthcare providers. This depth of analysis enriches our understanding of medication adherence and discontinuation, setting it apart from previous studies in the field. The detailed methodology employed to assess persistence, which takes into account variations in refill gaps and the intricacies of dosing regimens, further bolsters our comprehension of PSO treatment adherence. These collective strengths offer a nuanced and invaluable perspective on optimizing medication adherence in PSO management. This, in turn, holds the potential to inform tailored patient counseling strategies and, ultimately, enhance treatment outcomes. Importantly, our study makes a distinctive contribution by providing a more refined analysis of therapeutic changes, offering unique insights that advance the current understanding of medication adherence in psoriasis treatment compared to previous research.

## Figures and Tables

**Figure 1 pharmaceutics-15-02647-f001:**
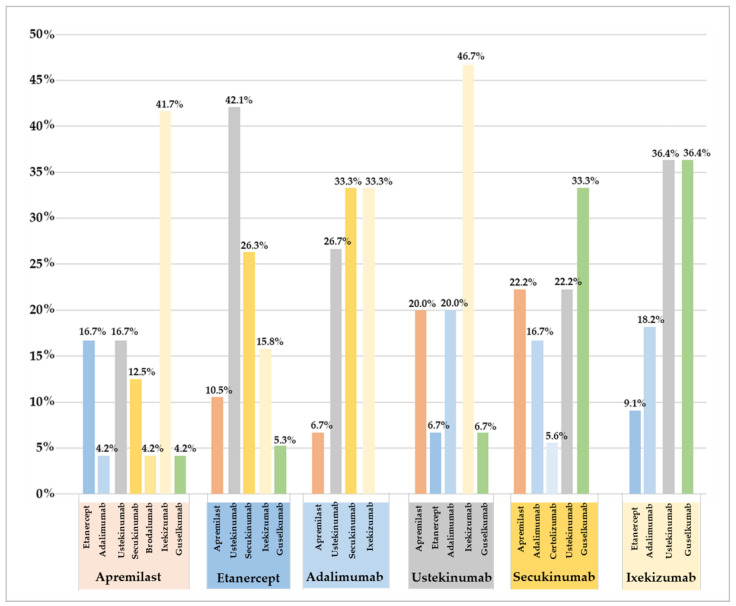
Swap therapy patterns. Notes: The figure depicts the percentages of patients who underwent therapy switching within the first year from the initiation of the index PSO therapy.

**Figure 2 pharmaceutics-15-02647-f002:**
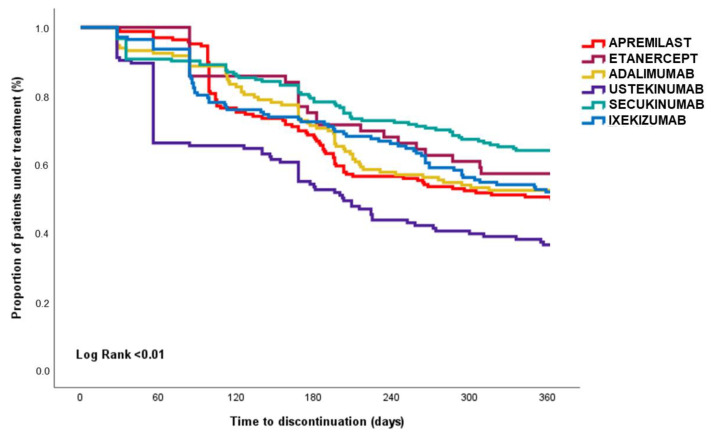
One year medication persistence rates to PSO treatments (gap 60). Notes: The figure illustrates the Kaplan–Meier curves, indicating the proportion (in percentage) of patients still under treatment during the first year following the initiation of the index PSO therapy (i.e., apremilast, etanercept, adalimumab, ustekinumab, secukinumab, and ixekizumab).

**Table 1 pharmaceutics-15-02647-t001:** Baseline patient’s characteristics stratified by PSO treatments.

	Overall °	Index Treatment					
		Apremilast	Etanercept	Adalimumab	Ustekinumab	Secukinumab	Ixekizumab
	n = 801	n = 165 (20.6%)	n = 56 (7.0%)	n = 132 (16.5%)	n = 124 (15.5%)	n = 183 (22.8%)	n = 141 (17.6%)
Gender, N (%)							
Males	485 (60.5)	91 (55.2)	33 (58.9)	77 (58.3)	70 (56.5)	122 (66.7)	92 (65.2)
Mean age ± SD	49.2 ± 16.3	60.9 ± 12.9	51.7 ± 14.5	37.0 ± 16.7	48.3 ± 14.9	48.9 ± 15.2	46.9 ± 13.8
Polypharmacy, * N (%)							
No polypharmacy (1–4 drugs)	258 (32.2)	34 (20.6)	22 (39.3)	47 (35.6)	38 (30.6)	69 (37.7)	48 (34.0)
Polypharmacy (5–9 drugs)	253 (31.6)	32 (19.4)	9 (16.1)	55 (41.7)	67 (54.0)	61 (33.3)	59 (41.8)
Excessive polypharmacy (≥10 drugs)	290 (36.2)	99 (60.0)	25 (44.6)	30 (22.7)	49 (39.5)	53 (29.0)	34 (24.1)

° Percentage calculated on the total of subjects analyzed. * Conditions occurring one year prior to the index date of initiation of psoriasis treatment. Abbreviations: SD, standard deviation.

**Table 2 pharmaceutics-15-02647-t002:** Adherence levels to PSO treatments.

1-Year Adherence Estimation	Overall	Apremilast	Etanercept	Adalimumab	Ustekinumab	Secukinumab	Ixekizumab
	n = 801	n = 165 (20.6%)	n = 56 (7.0%)	n = 132 (16.5%)	n = 124 (15.5%)	n = 183 (22.8%)	n = 141 (17.6%)
**Initiation Phase**							
∆% Ptp/Ptd	93.0	84.6	96.7	94.7	93.3	94.5	91.2
**Implementation phase**							
**Swap, N (%)**	106 (13.1)	24 (14.5)	19 (33.9)	15 (11.4)	15 (12.1)	18 (9.8)	11 (7.8)
Mean days ± SD	29 ± 84.8	34.3 ± 93.9	58.8 ± 98.6	24.5 ± 77.5	20.3 ± 64.8	27.2 ± 88.9	21.7 ± 79.2
**Switch, N (%)**	12 (1.5)	-	2 (3.6)	3 (2.3)	-	5 (2.7)	2 (1.4)
Mean days ± SD	4.1 ± 35.9	-	8 ± 49.9	7.4 ± 48.5	-	7.5 ± 49.1	3.9 ± 33.1
**Discontinuation phase**							
**30-day gap**							
Adherent, N (%)	390 (48.7)	78 (47.3)	29 (51.8)	66 (50.0)	40 (32.3)	110 (60.1)	67 (47.5)
Mean days ± SD	167.9 ± 107.1	163.9 ± 90	201.8 ± 110	164.9 ± 95.2	138.4 ± 109.3	182 ± 118.3	182.4 ± 115.5
**60-day gap**							
Adherent, N (%)	515 (64.3)	93 (56.4)	38 (67.9)	81 (61.4)	80 (64.5)	135 (73.8)	88 (62.4)
Mean days ± SD	176.6 ± 101.7	161.3 ± 75.4	222.7 ± 110.1	163 ± 90.5	177.9 ± 120.8	195.8 ± 120.3	176.4 ± 101.6
**90-day gap**							
Adherent, N (%)	570 (71.2)	99 (60.0)	42 (75.0)	89 (67.4)	102 (82.3)	142 (77.6)	96 (68.1)
Mean days ± SD	170 ± 90.9	153.1 ± 67.9	217.1 ± 103.3	152.7 ± 85.1	202.7 ± 95.4	183.5 ± 110	168.1 ± 93.9

Abbreviations: PSO, psoriasis; PTp, treatment plans prescribed; Ptd, treatment plans dispensed; SD, standard deviation.

**Table 3 pharmaceutics-15-02647-t003:** Predictors of non-persistence to PSO agents at 1-year post-initiation.

Characteristics	Unadjusted OR (95% CI)	*p*-Value	Adjusted OR (95% CI)	*p*-Value
Men (vs. women)	0.908 (0.686–1.203)	0.501	0.922 (0.681–1.249)	0.6
Age				
19–40 y (vs. under 18)	1.644 (0.734–3.681)	0.227	1.595 (0.663–3.838)	0.298
41–65 y (vs. under 18)	1.166 (0.533–2.549)	0.701	1.146 (0.472–2.783)	0.764
Over 65 y (vs. under 18)	1.5 (0.656–3.431)	0.337	1.585 (0.595–4.225)	0.357
Regimen complexity				
Polypharmacy (5–9 drugs) (vs. no polytherapy)	0.714 (0.505–1.010)	0.057	0.743 (0.515–1.072)	0.113
Excessive polypharmacy (≥10 drugs) (vs. no polytherapy)	0.76 (0.543–1.064)	0.11	0.679 (0.453–1.016)	0.06
Therapy change				
Swap	2.696 (1.723–4.219)	<0.001 *	2.835 (1.774–4.529)	<0.001 *
Switch	0.466 (0.139–1.561)	0.216	0.534 (0.152–1.875)	0.327
Index PSO treatment				
Apremilast	1.065 (0.756–1.499)	0.719	0.817 (0.222–3.003)	0.761
Etanercept	0.869 (0.505–1.496)	0.613	0.606 (0.154–2.382)	0.473
Adalimumab	0.932 (0.642–1.353)	0.71	0.712 (0.191–2.651)	0.612
Ustekinumab	2.226 (1.484–3.337)	<0.001 *	1.574 (0.423–5.861)	0.499
Secukinumab	0.546 (0.391–0.764)	<0.001 *	0.482 (0.132–1.761)	0.269
Ixekizumab	1.05 (0.73–1.51)	0.793	0.819 (0.222–3.023)	0.765

* *p*-value of less than 0.05 was considered to be statistically significant.

## Data Availability

All data used for the current study are available upon reasonable request to the Centro di Ricerca in Farmacoeconomia e Farmacoutilizzazione (CIRFF) authorized by the governance board of Unità del Farmaco della Regione Campania [D.G.R. 276, 23 May 2017].

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
