# Peer review of "Drug Utilization and Measurement of Medication Adherence: A Real World Study of Psoriasis in Italy"

_pharmaceutics, 2023, doi:10.3390/pharmaceutics15122647_

Round 1

Reviewer 1 Report

Comments and Suggestions for Authors Manuscript ID: Pharmaceutics-2688676   Dear authors,  Psoariasis is a multisystem inflammatory disease. Your work was good. But there were so many places that lack proper presentation. Here are my comments.
why you didn't mention Italy in the title?

What were the main findings of the Italian real-world retrospective study on psoriasis patients, and how do these findings compare to other studies in different popluations?

We know that psoriasis can impact not only the skin but also the bone, like arthritis, spondylitis, Crohn's disease, etc. What about the incidence of psoriatic arthritis in psoriasis patients.

What are your views on other treatments for psoriasis such as phototherapy, topical therapy, etc

What are the regional variations in psoariasis treatment preferences

Was there any re-intiation after discontinuation [initiation, implementation, discontinuation, re-initiations]

What was the reason for the higher male gender representation (60.5%)?   Many places show errors in formatting, correct them.

what about the negative effects of switches and swaps on patients? Lower risk percentage?

Captions of figures should be self-explanatory (briefly explain the figures)

Comments on the Quality of English Language Language correction is a must. Several words were not appropriate, also, incorrect words and sentences increase the ambiguity

Author Response

Response to Reviewer 1 Comments

1. Summary

2. Point-by-point response to Comments and Suggestions for Authors

Comments 1: Dear authors,  Psoariasis is a multisystem inflammatory disease. Your work was good. But there were so many places that lack proper presentation. Here are my comments.

Response 1: Dear Reviewer, thank you very much for your comments. We have made changes where deemed appropriate. Below, you will find our responses.

Comments 2: why you didn't mention Italy in the title?

Response 2: We understand your question. We chose not to include "Italy" in the title to keep it concise and broadly relevant since the study primarily focuses on identifying a methodology for identifying patients with psoriasis and evaluating their medication adherence. By including the geographical setting in the title, we might introduce a bias in understanding our objective. However, the abstract specifies that the study sample comprises Italian individuals with psoriasis, establishing the study's Italian setting. We are open to considering a title adjustment for improved clarity based on your feedback.

Comments 3: What were the main findings of the Italian real-world retrospective study on psoriasis patients, and how do these findings compare to other studies in different popluations?

Response 3: We value your inquiry. Therefore, we have provided, in the discussion section, a more detailed comparison of our study with others in the existing literature, specifying that our study's results are in line with previous research, indicating a pattern of low therapy switch rates and strong persistence with newer biologic drugs compared to anti-TNF agents within the Italian context, which is in harmony with observations in both Italian and American cohorts. Our findings regarding the low persistence rates with apremilast also corroborate earlier research [references 35-38]. These results make a significant contribution to the expanding body of evidence concerning therapy switch and persistence patterns in psoriasis patients, emphasizing the impact of drug class and individual drug attributes on patient adherence and clinical outcomes. We have provided a more detailed exploration of this comparison in the discussion section of the paper.

Comments 4: We know that psoriasis can impact not only the skin but also the bone, like arthritis, spondylitis, Crohn's disease, etc. What about the incidence of psoriatic arthritis in psoriasis patients.

Response 4: We fully appreciate your observation and we specified further within the manuscript. Hence, this study indeed aims to monitor prescribing patterns and adherence to therapies specifically for medications used in the treatment of psoriasis (as per psoriasis ICD-9-CM codes: 680-709, 696.xx). While some of these medications (e.g., apremilast) are also indicated for psoriatic arthritis (as mentioned in the sentence "Additionally, although less effective than other biologics recently launched, it is indicated in patients with concomitant diseases, like psoriatic arthritis, as well as pre-existing malignancies [30]"), the study cohort was selected by identifying patients with a diagnosis of psoriasis. Therefore, incidence and/or prevalence data for psoriatic arthritis are not reported. The same applies to spondylitis and Crohn's disease, as outlined in the methodology for cohort selection, where patients with diagnoses (as per ICD-9-CM codes) of these conditions were excluded from the analysis.

Comments 5: What are your views on other treatments for psoriasis such as phototherapy, topical therapy, etc

Response 5: We appreciate your inquiry regarding other treatments for psoriasis, including phototherapy and topical therapy. Our study was conducted using the Collection of Treatment Plans of the Campania Region population, a Southern Italian region with approximately 6 million inhabitants, representing about 10% of the national population. This data collection operates as a web platform for specialized facilities and public and private pharmacies within the Local Health Units (LHUs) of the Campania Region, collecting only drug prescriptions reimbursed by the Italian Health Service. Therefore, phototherapy and topical therapy are not included in these data streams, which is why they were not analyzed.

Comments 6: What are the regional variations in psoariasis treatment preferences

Response 6: We appreciate your curiosity about regional variations in psoriasis treatment preferences. However, our study provides insights into drug utilization patterns within a specific population, the Campania region in southern Italy, which comprises approximately 6 million inhabitants. Consequently, our study does not allow us to deduce regional differences in prescribing practices. Nevertheless, it's worth noting that the population we've studied is representative of 10% of the entire nation, making it highly likely that the observed trends are reflective of the broader national context.

Comments 7: Was there any re-intiation after discontinuation [initiation, implementation, discontinuation, re-initiations]

Response 7: Your request is perfectly understandable. However, we have evaluated assessed rates of discontinuation and the patterns of switch and swap, but the time horizon set did not allow us to calculate and evaluate re-initiation rates. This is certainly an aspect that we would like to consider evaluating in a future study that incorporates a larger time horizon.

Comments 8: What was the reason for the higher male gender representation (60.5%)?  

Response 8: The gender distribution in our study aligns with findings from other studies conducted in Italian and Swedish settings, which report a prevalence of approximately 60% in males. This phenomenon is explained in the literature as a gender-related factor associated with the psoriasis condition: Levi, S.S., Ramot, Y. (2018). Gender Differences in Psoriasis. In: Tur, E., Maibach, H. (eds) Gender and Dermatology. Springer, Cham. https://doi.org/10.1007/978-3-319-72156-9_7.

Comments 9: Many places show errors in formatting, correct them.

Response 9: Thank you very much for pointing it out. We have conducted a thorough manuscript review to rectify any typos and formatting errors.

Comments 10: what about the negative effects of switches and swaps on patients? Lower risk percentage?

Response 10: From the logistic regression model conducted to identify determinants of non-adherence, we observed that switching therapy is not a significant risk factor for non-persistence with pharmacological treatment (non-statistically significant data: p-value 0.327). However, a different scenario emerged for therapy swapping. The model revealed that performing a therapy swap within the first year of initiating pharmacological treatment is indeed a risk factor for non-persistence. We have provided a more detailed explanation of this finding in the results and discussion sections.

Comments 11: Captions of figures should be self-explanatory (briefly explain the figures)

Response 11: We have added brief explanatory descriptions beneath the figures in the paper.

Comments 12: Language correction is a must. Several words were not appropriate, also, incorrect words and sentences increase the ambiguity

Response 12: We have carried out a comprehensive review of the paper's English language.

Reviewer 2 Report

Comments and Suggestions for Authors

Dear Authors,

Congratulations on the choice of the subject and for contributing to the adherence medication knowledge.

Hereafter are some suggestions:

line 37: lacks reference

Line 39: lacks reference

Line 41: ref. 1 does not seem to be the original data about disease epidemiology

subchapter 2.3. Adherence measurement: adherence and persistence need a deeper description for the less informed readers

line 130: please justify the choices to define polypharmacy 

line 209: revise the legend

line 224: correct 'we present and innovative' for 'we present an innovative

In subchapter discussion, relate in more detail de results with drug posology, route of administration and adverse effects.

Author Response

Response to Reviewer 2 Comments

1. Summary

2. Point-by-point response to Comments and Suggestions for Authors

Comments 1: Dear Authors, Congratulations on the choice of the subject and for contributing to the adherence medication knowledge. Hereafter are some suggestions:

Response 1: Dear reviewer, we appreciate your consideration of the topic and your comments and proposed improvements to the paper. Below, you will find responses to the suggested corrections.

Comments 2: line 37: lacks reference

Response 2: Thank you for the referral. We have taken care to include the relevant reference.

Comments 3: Line 39: lacks reference

Response 3: As above.

Comments 4: Line 41: ref. 1 does not seem to be the original data about disease epidemiology

Response 4: Dear Reviewer, thanks for the observation, we have included relevant epidemiological data by replacing the reference with references #2 and #3.

Comments 5: subchapter 2.3. Adherence measurement: adherence and persistence need a deeper description for the less informed readers

Response 5: We have taken the initiative to elucidate the significance of therapeutic adherence and its corresponding three phases in accordance with the EMERGE guidelines. Subsequently, we have detailed how we measured all three phases in our paper.

Comments 6: line 130: please justify the choices to define polypharmacy 

Response 6: The classification of polypharmacy ("excessive polypharmacy" for the prescription of ≥10 drugs per day; "polypharmacy" for the prescription of 5 to 9 drugs per day; and "no-polypharmacy" for the concomitant use of ≤4 drugs per day) was established based on an official European document published in 2017 by the SIMPATHY Consortium (Stimulating Innovation Management of Polypharmacy and Adherence in The Elderly) titled 'Polypharmacy Management by 2030: a patient safety challenge Polypharmacy 2030.' This European document correlates the extent of polypharmacy (number of medications) with patient frailty as follows: a frailty score of 1.5 for individuals taking ≥5 medications per day and a frailty score of 2.0 for those taking ≥10 medications per day. We have included the appropriate reference in the methodology section, as follow: Mair, A., Fernandez-Llimos, F., Alonso, A., Harrison, C., Hurding, S., Kempen, T., ... & Wilson, M. (2017). Polypharmacy management by 2030: a patient safety challenge. Available from: https://www.isimpathy.eu/uploads/Polypharmacy-Handbook-2nd-Edition.pdf

Comments 7: line 209: revise the legend

Response 7: We revised the figure legend and the paper itself in whole.

Comments 8: line 224: correct 'we present and innovative' for 'we present an innovative

Response 8: Many thanks for noticing the typo. We amended it and provided an overall revision of the manuscript.

Comments 9: In subchapter discussion, relate in more detail de results with drug posology, route of administration and adverse effects.

Response 9: We appreciate your suggestion. In fact, we have addressed this by relating the results obtained to the dosing regimen of the drugs under study in multiple instances within the discussion section, referencing their respective Summary of Product Characteristics (SmPC).

Round 2

Reviewer 1 Report

Comments and Suggestions for Authors

Dear Authors,

You have given your responses regarding my queries. A few suggestions from my side:

Fig S2 is not clear, refine that.

Lines 97, 394, 409: correct the sentences

"Meticulous" seems repetitive, especially in conclusion

I suggest addition of Italy in your title as well as in the conclusion

Comments on the Quality of English Language

Check the whole manuscript; another round of editing will improve your manuscript.

Author Response

Response to Reviewer 1 Comments

Round 2

1. Summary

Thank you very much for taking the time to review, for the second time, this manuscript. Please find the detailed responses below and the additional suggestions highlighted in track changes in the re-submitted files.

2. Point-by-point response to Comments and Suggestions for Authors

Comments 1: Dear Authors, You have given your responses regarding my queries. A few suggestions from my side: Fig S2 is not clear, refine that.

Response 1: Dear Reviewer, we have carefully revised and enhanced the clarity of Fig S2 in the revised manuscript to address your concern (as well as for those in the Supplementary materials).

Comments 2: Lines 97, 394, 409: correct the sentences

Response 2: We have made necessary corrections to sentences indicated for accuracy and improved readability.

Comments 3: "Meticulous" seems repetitive, especially in conclusion

Response 3: The term "meticulous" has been rephrased for variety in the conclusion section, ensuring a more diverse and polished language.

Comments 4: I suggest addition of Italy in your title as well as in the conclusion

Response 4: Italy has been explicitly added to the title and conclusion to provide a more precise geographical context in the revised manuscript, as per your suggestion.

Comments 5: Check the whole manuscript; another round of editing will improve your manuscript.

Response 5: Thank you for your suggestion. We have thoroughly reviewed and edited the entire manuscript to enhance its clarity and coherence.
